



# Phenomena preceding major earthquakes interconnected through a physical model

Panayiotis A. Varotsos[1,2], Nicholas V. Sarlis[1,2], and Efthimios S. Skordas[1,2]

[1]Section of Solid State Physics, Department of Physics, National and Kapodistrian University of Athens, Panepistimiopolis, Zografos 157 84, Athens, Greece
[2]Solid Earth Physics Institute, Department of Physics, National and Kapodistrian University of Athens, Panepistimiopolis, Zografos 157 84, Athens, Greece

**Correspondence:** Nicholas V. Sarlis (nsarlis@phys.uoa.gr)

**Abstract.** The analysis of earthquake time series in a new time domain termed natural time enables the uncovering of hidden properties in time series of complex systems and has been recently employed as basis of a method to estimate seismic risk. Applying this analysis, as an example, to the Japanese seismic data from 1 January 1984 until the super-giant M9 Tohoku earthquake on 11 March 2011, we find that almost three months before its occurrence the entropy change of seismicity under time

reversal is minimized on 22 December 2010, which signals an impending major earthquake. On this date the order parameter fluctuations of seismicity exhibit abrupt increase. This increase is accompanied by various phenomena, e.g., from this date the horizontal GPS azimuths start to become gradually oriented toward the southern direction, while they had random orientation during the preceding period. Two weeks later, a minimum of the order parameter fluctuations of seismicity appears accompanied by anomalous Earth's magnetic field variations and by full alignment of the orientations of GPS azimuths southwards

leading to the most intense crust uplift. These phenomena are discussed and found to be in accordance with a physical model which seems to explain on a unified basis anomalous precursory changes observed either in ground-based measurements or in satellite data.

## 1 Introduction

Almost eight years have passed since the Tohoku earthquake (EQ) that occurred on 11 March 2011 in Japan with magnitude (M) 9.0, the largest magnitude ever recorded in Japan. In the meantime, independent research groups reported anomalous precursory variations of quantities like the geomagnetic field (Xu et al., 2013; Han et al., 2015), seismicity upon analyzing it in natural time (Sarlis et al., 2013, 2015; Varotsos et al., 2014) and Earth's surface displacements measured by Global Positioning System (GPS) (Chen et al., 2014). In the 1980's, a short term earthquake prediction method was introduced based on the observation

of Seismic Electric Signals (SES), which are low frequency transient changes of the electric field of the Earth preceding EQs (Varotsos and Alexopoulos, 1984a, b; Varotsos et al., 1986). Several SES recorded within a short time are termed SES activity



(Varotsos and Lazaridou, 1991). Major EQs are preceded by intense SES activities accompanied by clear Earth's magnetic field variations (Varotsos et al., 2003b) mainly recorded on the z-component (Sarlis and Varotsos, 2002). This method was motivated by a condensed matter physical model for SES generation, which also foresees that some additional transient multidisciplinary phenomena should be simultaneously generated and ended well before the EQ rupture as schematically shown in Fig. 1(a).

This was in direct contrast to other proposed precursory mechanisms (Rikitake, 1981) that usually exhibit anomalous behavior becoming more intense upon approaching the EQ failure as seen in Fig. 1(b). The scope of the present study is twofold: First, investigate whether the transient phenomena suggested by this model actually appeared before the M9 Tohoku EQ, having been observed by ground based measurements or by GPS data. Second, to report other precursory phenomena that appeared before the Tohoku EQ almost simultaneously with the transient phenomena that were expected on the basis of this model.

The aforementioned condensed matter physical model for SES generation, termed pressure stimulated polarization currents (PSPC) model (Varotsos and Alexopoulos, 1986; summarized also in Varotsos and Alexopoulos, 1984a, b, and Varotsos et al., 1993), suggests the following (see Fig. 2): In the Earth, electric dipoles always exist (Varotsos and Alexopoulos, 1986) due to lattice imperfections (point and linear defects (e.g., see Varotsos, 2008)) in the ionic constituents of rocks. In the future focal region of an EQ, where the electric dipoles have initially random orientations (Fig. 2c), the stress, $\sigma$, starts to gradually increase

due to an excess stress disturbance (Fig. 2a). Let us call hereafter this stage A. When this gradually increasing stress reaches a critical value ($\sigma_{cr}$), the electric dipoles exhibit a cooperative orientation (Fig. 2e) resulting in the emission of a transient electric signal SES (Fig. 2b) with current density $j$. We call this stage B. It has been pointed out by Uyeda et al. (2009b) that the PSPC model is unique among other models in that SES would be generated spontaneously during the gradual increase of stress without requiring any sudden change of stress such as microfracturing (Molchanov and Hayakawa, 1995) (or faulting).

This model is in contrast to other physical mechanisms, like the electrokinetic phenomena, suggested also for the explanation of SES generation (Varotsos, 2005).

 Observations of SES activities in Japan (Uyeda et al., 2000, 2002, 2009a), in China ( see Huang, 2011 and references therein, for example see the geoelectric field changes depicted in Fig. 2b of Fan et al., 2010 that started almost 50 days before the Ms8.0 Wenchuan EQ in 2008) as well as in Mexico (see p. 220 of Ramírez-Rojas et al., 2011) and in California (where magnetic

field variations similar to those associated with the SES activities in Greece have been reported, e.g., see Fraser-Smith et al., 1990; Bernardi et al., 1991; see also Varotsos et al., 2011b) have shown that their lead time is from a few weeks to around $5\frac{1}{2}$ months or so, in agreement with earlier observations in Greece Varotsos and Lazaridou, 1991; Varotsos et al., 1993). Hence, the SES observations in various EQ prone areas reveal (Varotsos et al., 2011b) that before the occurrence of major EQs there is a crucial time scale (from a few weeks to) around a few months or so, in which the critical stress $\sigma_{cr}$ is attained and changes

in other associated physical quantities should become also detectable (Varotsos et al., 2011a).

 This paper is structured as follows: In the next section, i.e., Section 2, we present the anomalous variations of multidisciplinary nature observed before the M9 Tohoku EQ by independent research groups, while our own findings obtained by natural time analysis of the seismicity of Japan are given in Section 3. In the subsequent Section 4, we investigate whether the observed precursory variations are in accordance with the PSPC model and in the final section, i.e., Section 5, we summarize

our conclusions.



## 2   Anomalous variations of multidisciplinary nature observed by independent groups before the 2011 M9 Tohoku EQ.

### 2.1   Earth's magnetic field variations

Xu et al. (2013) found anomalous behavior of geomagnetic diurnal variations mainly in the vertical component at the Esashi station (ESA) located at about 135km from the M9 Tohoku EQ focal zone for about 10 days, i.e., 4-14 January 2011. They
analyzed geomagnetic data of a 3-year period, i.e., from 1 January 2010 to 31 December 2012, by computing ratios of diurnal variation range between the target station ESA and the remote reference station Kakioka about 300 km from the EQ epicenter. To validate this finding, further investigations have been reported by Han et al. (2015) after analyzing geomagnetic data of 16 years' long term observations in Japanese stations. They again found that the mean values of the ratios of the diurnal variations in the vertical component showed a clear anomaly exceeding the statistical threshold during the aforementioned period, i.e., 4-
14 January 2011, and in addition they emphasized that this anomaly is unique in over 16 years. This has been further validated by the most recent study of Han et al. (2016) who analyzed geomagnetic data of long-term observations at 17 stations in Japan. They found that the above unique anomaly in the vertical component has been also observed at a second station at Mizusawa (MIZ) in the Tohoku region, which is about 20km southwest to the ESA station. This fact that both ESA and MIZ show clear anomalies at the same time suggests that the anomaly cannot be resulted from observation system error or artificial noises (Han
et al., 2016). Furthermore, this anomaly cannot be attributed to magnetic storms since it has been observed during a period in which no moderate-strong magnetic storms have been recorded.

### 2.2   Earth's surface displacements

Daily resolution data retrieved from the 1243 GPS stations in Japan have been utilized by Chen et al. (2014) to expose surface displacements before the M9 Tohoku EQ. They applied the method proposed by Chen et al. (2011) on filtering long-term plate
movements, short-term noise and frequency dependent (i.e., semi-annual and annual) variations from the 3-component GPS data for all stations. The NS and EW components were utilized to compute the orientations of the horizontal azimuths, termed GPS azimuths.

In general, the residual surface displacements are random (Chen et al., 2014, 2011). However, as depicted by black letters in Fig. 3, southward movements became evident (Chen et al., 2014) on 5 January 2011, i.e., 65 days before the Tohoku EQ. Other
changes before and after 5 January 2011 have been also observed as follows: While during the period 12-22 December 2010 random orientations of GPS azimuths prevailed, a gradual alignment toward the southern direction started on 22 December 2010 and continued until 5 January 2011 accompanied with a gradual uplift of the crust. The most intense crust uplift was observed on 5 January 2011 together with the full alignment of GPS azimuths southwards. Additional details will be given in the Section 4.





## 2.3 Changes of the level and temperature of confined groundwater.

Orihara et al. (2014) reported that anomalous groundwater changes started three months before the Tohoku EQ. In particular, groundwater level and temperature decreased almost simultaneously in a 2000 m well at a spa, Goyo-onsen, in Iwate Prefecture, 155 km northwest of the epicenter. This simultaneous decrease occurred only once in 3.5 year records since the

recordings started in this source since October 2007. Orihara et al. (2014) emphasized that Tohoku EQ is the only EQ that was preceded by anomalous changes in both water level and temperature. The exact date of the initiation of this phenomenon is not mentioned explicitly by Orihara et al. (2014), because as they state the measurements were not made continuously but were taken intermittently and irregularly (the average interval between them being 8 days). They plotted, however, the consecutive measurements versus conventional time in their Fig. 1 (period 2007-2012) in which one can read that the phenomenon initiated

around 20 December 2010, which agrees with what they state, i.e., around 3 months before Tohoku EQ.

In addition, Orihara et al. (2014) reported that according to radon concentration measurements (Tsunomori and Tanaka, 2014) in the ground water in the Izu peninsula (at a distance about 500 km from the epicenter), an increase started almost three months before Tohoku EQ. This increase occurred only this time during a 35-year observation.

## 3 Precursory changes observed by means of natural time analysis of Japanese seismicity.

Natural time analysis uncovers important hidden properties in time series of complex systems (Varotsos et al., 2011b) and has been recently employed by Turcotte and coworkers as basis of a new methodology to estimate the current seismic risk level (Rundle et al., 2016, 2018; Luginbuhl et al., 2018a, b).

### 3.1 Natural time analysis. Background.

In a time series comprising $N$ EQs, the natural time for the occurrence of the $k$-th EQ of energy $Q_k$ is defined as $\chi_k = k/N$.

In natural time analysis, we study the evolution of the pair $(\chi_k, p_k)$, where

$$p_k = Q_k / \sum_{n=1}^{N} Q_n \tag{1}$$

denotes the normalized energy released during the $k$-th EQ. $Q_k$ and hence $p_k$ for earthquakes is estimated through the relation (Kanamori, 1978)

$$Q_k \propto 10^{1.5 M_k} \tag{2}$$

It is widely accepted (Carlson et al., 1994; Holliday et al., 2006) that the observed earthquake scaling laws indicate the existence of phenomena closely associated with the proximity of the system to a critical point. In particular, it has been indicated by Carlson et al. (1994) that it seems possible that systems that operate persistently near a threshold of instability are in some way like thermodynamic systems near critical points (EQ can be regarded as a stick-slip frictional instability of a





pre-existing fault). The order parameter of seismicity is the quantity by which one can identify the approach of the dynamical system to a critical point. It was argued by Varotsos et al. (2005c) that the variance

$$\kappa_1 = \langle \chi^2 \rangle - \langle \chi \rangle^2 \tag{3}$$

of natural time $\chi$ weighted for $p_k$, given by

$$\kappa_1 = \sum_{k=1}^{N} p_k (\chi_k)^2 - \left( \sum_{k=1}^{N} p_k \chi_k \right)^2, \tag{4}$$

may serve as an order parameter of seismicity.

The entropy $S$ in natural time is defined (Varotsos et al., 2003a, 2004) by

$$S \equiv \langle \chi \ln \chi \rangle - \langle \chi \rangle \ln \langle \chi \rangle \tag{5}$$

where the brackets $\langle \ldots \rangle \equiv \sum (\ldots) p_k$ denote averages with respect to the distribution $p_k$, i.e., $\langle f(\chi) \rangle \equiv \sum f(\chi_k) p_k$. Notably, the functional given by Eq.(5) has been shown (Varotsos et al., 2005b) to exhibit positivity, concavity, and experimental stability, which are the three requirements in order to be characterized as entropic functional. Furthermore, note that the entropy $S$ is a dynamic entropy (Varotsos et al., 2004) depending on the sequential order of the events and not simply a statistical entropy (e.g. Shannon entropy), (see Varotsos et al., 2005a). Upon considering time reversal $\widehat{T}$, i.e., $\widehat{T} p_k = p_{N-k+1}$, the value $S$ changes to a value $S_-$:

$$S_- = \sum_{k=1}^{N} p_{N-k+1} \left( \frac{k}{N} \right) \ln \left( \frac{k}{N} \right) - \left( \sum_{k=1}^{N} \frac{k}{N} p_{N-k+1} \right) \ln \left[ \sum_{l=1}^{N} \frac{l}{N} p_{N-l+1} \right] \tag{6}$$

The physical meaning of the entropy change $\Delta S \equiv S - S_-$ in natural time under time reversal is discussed in Varotsos et al. (2007, 2011b).

Using a moving window of length $i$ (number of events) sliding through the time series of $L$ consecutive events the entropy in natural time is determined for each position $j = 1, 2, \ldots, L - i$ of the sliding window. Thus, a time series of $S_i$ is obtained. By considering the standard deviation $\sigma(\Delta S_i)$ of the time series of $\Delta S_i \equiv S_i - (S_-)_i$, we define (Varotsos et al., 2011b; Ramírez-Rojas et al., 2018; Sarlis et al., 2018b) the complexity measure $\Lambda_i$

$$\Lambda_i = \frac{\sigma(\Delta S_i)}{\sigma(\Delta S_{100})} \tag{7}$$

where the denominator has been selected (Ramírez-Rojas et al., 2018) to correspond to the standard deviation $\sigma(\Delta S_{100})$ of the time series of $\Delta S_i$ of $i=100$ events.

$\Delta S$ constitutes a key measure that may identify (Varotsos et al., 2011b) when the system approaches the critical point (dynamic phase transition). For example, $\Delta S$ has been applied for the identification of the time of an impending sudden cardiac death risk (Varotsos et al., 2007). Furthermore, it has been used (Sarlis et al., 2011) for the study of the predictability



of the Olami-Feder-Christensen (OFC) model for earthquakes (Olami et al., 1992), which is probably (Ramos et al., 2006) the most studied non-conservative self-organized criticality (SOC) model. The OFC model originated by a simplification of the Burridge and Knopoff spring-block model (Burridge and Knopoff, 1967) by mapping it into a non-conservative cellular automaton simulating the earthquake's behavior and introducing dissipation in the family of SOC systems. In particular, it

was found that $\Delta S$ exhibits a clear minimum (Varotsos et al., 2011b) (or maximum if we define (e.g., see Sarlis et al., 2011) $\Delta S \equiv S_- - S$ instead of $\Delta S \equiv S - S_-$) before a large avalanche in the OFC model, which corresponds to a large earthquake. For example, by analyzing the seismicity during the period 2012-2017 in natural time in the Chiapas region of Mexico where the M8.2 earthquake occurred on 7 September 2017, we observed (Sarlis et al., 2018b) that the entropy change $\Delta S$ of seismicity under time reversal was minimized almost three months before and in particular on 14 June 2017.

## 3.2 Results from natural time analysis of seismicity.

Interesting results have been recently obtained upon analyzing the Japan seismic catalogue in natural time and computing the fluctuations of $\kappa_1$. To compute the $\kappa_1$ fluctuations, the procedure explained in Sarlis et al. (2013, 2015) and Varotsos et al. (2014) was applied by using a sliding natural time window comprising the number $i$ of EQs that would occur on average in a few months, or so. We then calculate the average value $\mu(\kappa_1)$ and the standard deviation $\sigma(\kappa_1)$ of the ensemble of $\kappa_1$ obtained.

The quantity

$$\beta_i \equiv \sigma(\kappa_1)/\mu(\kappa_1) \tag{8}$$

is defined as the variability of $\kappa_1$ (Varotsos et al., 2011b). The time evolution of the $\beta$ value can then be pursued by sliding the excerpt $i$ through the EQ catalogue and the corresponding minimum value is labeled $\beta_{min}$. The following key-results have been obtained:

Varotsos et al. (2013) found that the fluctuations of the order parameter $\kappa_1$ of seismicity exhibited a clearly detectable minimum approximately at the time of the initiation of a pronounced SES activity recorded by Uyeda and coworkers (Uyeda et al., 2002, 2009a) around 2 months before the volcanic-seismic swarm activity in 2000 in the Izu Island region, Japan.

Sarlis et al. (2013) analyzed the Japan seismic catalog in natural time from 1 January 1984 to 11 March 2011 and their results showed that the fluctuations $\beta$ of the order parameter of seismicity exhibited distinct minima $\beta_{min}$ a few months before all the

shallow earthquakes of magnitude 7.6 or larger that occurred during this 27-year period in the Japanese area $N_{25}^{46}E_{125}^{148}$. Among these minima, the minimum before the M9 Tohoku EQ observed at around 5 January 2011 was the deepest. Subsequently, Sarlis et al. (2015) found that the spatiotemporal variations of $\beta_{min}$ enable the estimation of the epicentral area of the impending mainshock for all these EQs of magnitude 7.6 or larger.

In addition, Varotsos et al. (2014) focused on the minima $\beta_{min}$ preceding all EQs of magnitude 8 (and 9) class that occurred

in the Japanese area from 1 January 1984 to 11 March 2011 and applied Detrended Fluctuation Analysis (DFA) (Peng et al., 1993, 1994) to the earthquake magnitude time series. DFA has been established as a standard method to investigate long range correlations in non-stationary time series in diverse fields (e.g., see Peng et al., 1993, 1994; Telesca and Lovallo, 2009) including the study of geomagnetic data associated with the M9.0 Tohoku EQ (Rong et al., 2012). The results of DFA are



described in terms of the so called DFA exponent (Peng et al., 1993, 1994) hereafter labeled $\alpha$ ($\alpha = 0.5$ means random, $\alpha$ greater than 0.5 long-range correlations, and $\alpha$ less than 0.5 anti-correlations). The following three main features have been identified (Varotsos et al., 2014): The minima $\beta_{min}$ are observed during periods when long range temporal correlations between EQ magnitudes have been developed since the corresponding DFA exponent is $\alpha > 0.5$. Before (*bef*) the minima $\beta_{min}$ there

exists a stage in which an evident anti-correlated behavior appears showing a minimum $\alpha_{min,bef}$ in the DFA exponent markedly smaller than 0.5. Finally, after (*aft*) the minima $\beta_{min}$, the long range correlations break down to an almost random behavior ($\alpha = 0.5$) possibly turning to anti-correlation exhibiting a minimum $\alpha_{min,bef}$ with $\alpha_{min,bef} < 0.5$. These three main features of temporal correlations between EQ magnitudes can be visualized in Fig. 3 for the case of M9 Tohoku EQ. The deepest minimum $\beta_{min}$ was observed around 5 January 2011 during a period in which long range correlations between EQ magnitudes

prevail with a DFA exponent $\alpha > 0.5$. Before this minimum an anti-correlated behavior was identified on 22 December 2010 with $\alpha_{min,bef} = 0.35$. After $\beta_{min}$ the long-range correlations break down on 13 January 2011 to an almost random behavior with $\alpha = 0.5$ and subsequently the behavior turned to anti-correlation on 23 January 2011 with $\alpha_{min,aft} = 0.42$. As it will be commented in the next section, the main features of the temporal correlations between EQ magnitudes obtained by DFA appeared simultaneously with distinct phases of crustal deformation identified by GPS measurements described in the previous

section.

Recently we have shown that almost three months before the M9 Tohoku earthquake, i.e., on 22 December 2010, the following additional facts have been observed: First, the complexity measure $\Lambda_i$ associated with the fluctuations of the entropy change of seismicity under time reversal exhibited abrupt increase which conforms to the seminal work by Lifshitz and Slyozov (Lifshitz and Slyozov, 1961) and independently by Wagner (Wagner, 1961) for phase transitions showing that the characteristic

size of the minority phase droplets exhibits a scaling behavior in which time growth has the form $A(t - t_0)^{1/3}$ (Varotsos et al., 2018). It was also found that the increase $\Delta\Lambda_i$ of $\Lambda_i$ follows the latter form and that the prefactors $A$ are proportional to the scale $i$, while the exponent $(1/3)$ is independent of $i$ (Varotsos et al., 2018). Second, the Tsallis entropic index $q$ (Tsallis, 1988), shows a simultaneous increase which interestingly exhibits the same exponent $(1/3)$ but the prefactors $A$ are not proportional to the scale $i$ (Varotsos et al., 2018). Third, a minimum $\Delta S_{min}$ of the change $\Delta S$ of the entropy of seismicity in the entire

Japanese region under time reversal was found by Sarlis et al. (2018a), who also demonstrated that the probability to obtain such a minimum by chance is approximately 3% thus showing that it is statistically significant. In addition, the robustness of the appearance of this minimum on 22 December 2010 upon changing the EQ depth, the EQ magnitude threshold, and the size of the area investigated has been documented (Sarlis et al., 2018a). Such a minimum is of precursory nature signaling that a large EQ is impending according to the natural time analysis of the OFC model as mentioned in subsection 3.1. Fourth,

studying the fluctuations $\beta$ of the order parameter of seismicity in the entire Japanese region $N_{25}^{46}E_{125}^{148}$ versus the conventional time from 1 January 1984 until the Tohoku EQ occurrence on 11 March 2011, we find (Varotsos et al., 2019) a large fluctuation of $\beta$ upon the occurrence of the M7.8 near Chichi-jima EQ on 22 December 2010. This finding has been also checked for several lengths from $i =$150 to 500 events, which also revealed the following (Varotsos et al., 2019): Upon increasing $i$ it is observed (see Figs. 2B and 4E of Sarlis et al., 2013) that the increase $\Delta\beta_i$ of the $\beta_i$ fluctuation on 22 December 2010 becomes

distinctly larger -obeying the interrelation $\Delta\beta_i = 0.5\ln(i/114.3)$- which however does not happen (see Figs. 4A-4D of Sarlis





et al., 2013)] for the increases of the $\beta$ fluctuations upon the occurrences of all other shallow EQs in Japan of magnitude 7.6 or larger during the period from 1 January 1984 to the time of the M9 Tohoku EQ. Hence, the $\beta$ fluctuation on 22 December 2010 accompanying the minimum $\Delta S_{min}$ is unique. Its presence is of paramount importance for the check of the validity of the physical model that will be discussed in the next Section.

## 5   4   Compatibility of the observed precursory phenomena with the PSPC physical model

Let us now discuss the multidisciplinary observations described in the previous two sections (and compiled in Fig. 3) that preceded the M9 Tohoku EQ. As we shall see all these observations are directly evidenced from the PSPC model, except probably of the anomalous changes of the level and temperature of confined groundwater.

A striking fact is that on 5 January 2011 the phenomenon of aligned orientations of the GPS azimuths occurred almost
simultaneously with two other phenomena, i.e., the initiation of the anomalous Earth's magnetic field variations and the minimum $\beta_{min}$ of the fluctuations of the order parameter of seismicity. This is strikingly reminiscent of the mechanism of the emission of SES activity (stage B of PSPC model) in which upon reaching $\sigma = \sigma_{cr}$, the electric dipoles exhibit cooperative orientation that also reflects alignment of the horizontal GPS azimuths (black in Fig. 3), as expected by Varotsos et al. (2011a), leading to the most intense crust uplift. Such an SES emission is directly evidenced by the observed anomalous variations of
the Earth's magnetic field mainly in the vertical component (orange in Fig. 3). This emission is consistent with the observation of the deepest minimum $\beta_{min}$ on 5 January 2011 of the fluctuations of the order parameter of seismicity (blue in Fig. 3) in view of the up to date experimental results mentioned above that an SES activity initiates almost simultaneously with both the observation of $\beta_{min}$ and the establishment of long range temporal correlation between earthquake magnitudes.

Another striking fact is that the aforementioned simultaneous appearance of the three phenomena around 5 January 2011
has been preceded by a stage of an evident anti-correlated behavior between earthquake magnitudes since it was found that $\alpha_{min,bef} = 0.35$ upon the occurrence on 22 December 2010 of the M7.8 EQ in southern Japan at $27.05^{o}$N $143.94^{o}$E. On the same date, the horizontal GPS azimuths, which were initially random, started to become gradually oriented toward the southern direction probably due to an excess stress disturbance. This may also originate the simultaneous appearance of various phenomena including the large abrupt increase of the order parameter fluctuations along with an abrupt increase of the complexity
measure $\Lambda_i$ of the change of the entropy of seismicity under time reversal (cf. recall that this change is then minimized), an increase of the Tsallis entropic index $q$ (Tsallis, 1988) and start of groundwater anomalous changes, i.e., groundwater level drop, temperature decrease and increase of radon concentration. This corresponds to the stage A of the PSPC model according to which an excess stress disturbance starts gradually increasing until reaching $\sigma_{cr}$.

After the occurrence of $\beta_{min}$ at around 5 January 2011, the intense crust uplift was gradually mitigated and the orientations
of GPS azimuths returned (Chen et al., 2014) to random around 13 January 2011, thus agreeing with the DFA exponent $\alpha = 0.5$ (Varotsos et al., 2014). The behavior turned to anti-correlation around 23 January 2011 with DFA exponent $\alpha_{min,aft} = 0.42$ and a shift of earthquake-related stress disturbance was observed (Chen et al., 2014) where westward movements replaced the southward ones, i.e., the orientations of the residual displacements were re-aligned along the western direction and the



crust depressed. After this change on 23 January 2011 the stress disturbance gradually approached the threshold of the fault rupture and the orientations of the residual displacements became random again (Chen et al., 2014) in agreement with the DFA exponent of the earthquake magnitude time series being close to 0.5 until around 10 February 2011 (see Fig. 5 of Varotsos et al., 2014), which indicates random behavior. This fact that the Tohoku EQ occurred after the emergence of an almost random behavior did not come as a surprise since it is strikingly reminiscent of similar findings in other complex time series as follows: In the case of electrocardiograms, for example, the long-range temporal correlations that characterize the healthy heart rate variability break down for individuals at high risk of sudden cardiac death (SCD), and this is often accompanied by emergence of uncorrelated randomness (Varotsos et al., 2011b; Goldberger et al., 2002) (SCD could be viewed as a critical phenomenon, e.g., see Varotsos et al., 2004, 2005a, 2007).

## 5   Summary and Conclusions

Several phenomena of multidisciplinary nature preceded the M9 Tohoku EQ that occurred on 11 March 2011. Leaving aside the details, these phenomena were mainly accumulated around two dates, i.e., 22 December 2010 and 5 January 2011, which concur with the two stages A and B of the PSPC physical model, respectively. These phenomena include:

A) Around the date 22 December 2010:

1. The entropy change of seismicity under time reversal is minimized along with increased fluctuations (since $\Lambda_i$ increases).

2. Increase of the fluctuations of the order parameter of seismicity.

3. The DFA exponent decreased to the value $\alpha_{min,bef} = 0.35$, which is the lowest observed during the period 1984-2011 of our study, pointing to an evident anti-correlated behavior in the earthquake magnitude time series.

4. The horizontal GPS azimuths started to become gradually oriented toward the southern direction (while they had random orientations during the preceding period 12-22 December 2010).

5. Anomalous changes of the groundwater started (level drop, temperature decrease, and probably increase of radon concentration).

6. Increase of Tsallis entropic index $q$.

B) Around the date 5 January 2011:

1. Unprecedented minimum $\beta_{min}$ of the fluctuations of the order parameter of seismicity.

2. Anomalous magnetic field variations started (which reflects the initiation of a strong SES activity).

3. Full alignment of the orientations of the GPS azimuths southwards accompanied by the most intense crust uplift.

4. Long range temporal correlations in the earthquake magnitude time series.

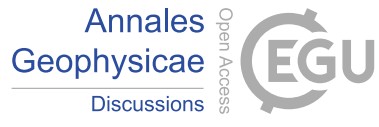

All the above phenomena were observed to begin and end well before the M9 Tohoku EQ occurrence as schematically shown in Fig. 1(a) in accordance with the PSPC model (Fig. 2).

*Author contributions.*  P.A.V., N.V.S. and E.S.S. designed research, performed research, analysed the results, and reviewed the manuscript.

*Competing interests.*  All authors declare no competing financial and non-financial interests.





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





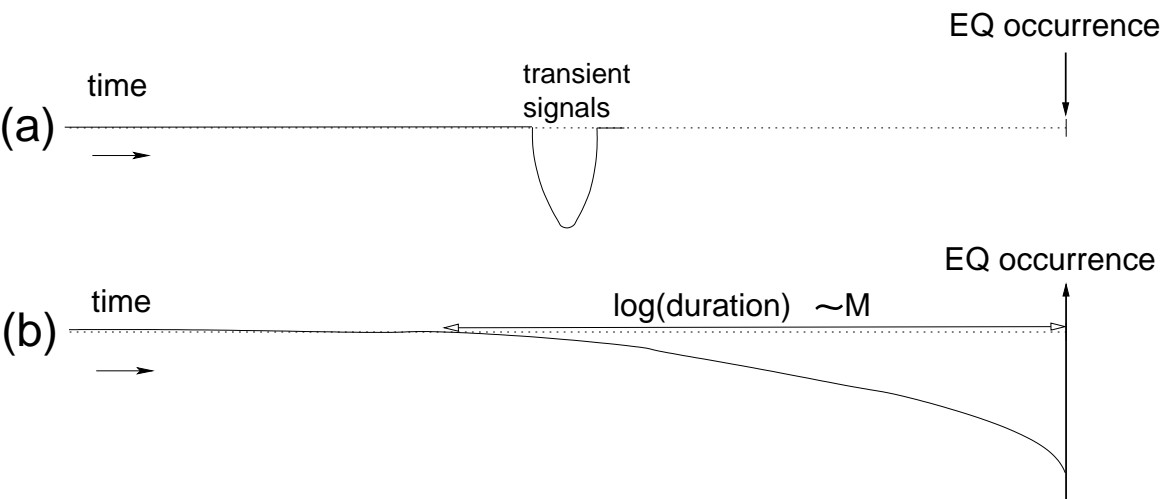

**Figure 1.** Schematic diagram showing the two distinct approaches proposed for a precursory behavior. (a): The case of the condensed matter physical model for SES generation, which differs greatly from other suggested mechanisms (Rikitake, 1981) in which the anomalous precursory behavior becomes more intense upon approaching the EQ occurrence (b).


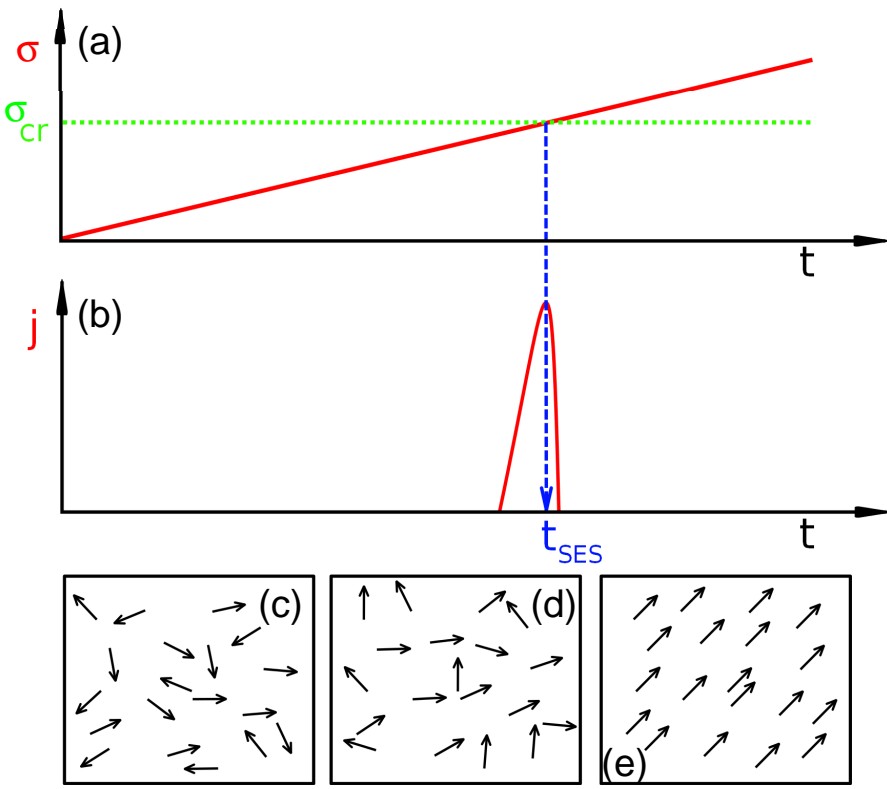

**Figure 2.** Schematic diagram of the condensed matter physical model proposed for the SES generation. (a) Before an EQ, the stress $\sigma$ gradually increases in the focal area versus the time t towards reaching a critical value $\sigma_{cr}$. (b) When $\sigma$ reaches $\sigma_{cr}$ a transient electric signal is emitted that constitutes an SES. (c) Random orientation of the electric dipoles at small stress. (d) Partial orientation at an intermediate stress $\sigma(< \sigma_{cr})$. (e) Cooperative orientation of the electric dipoles when $\sigma = \sigma_{cr}$.



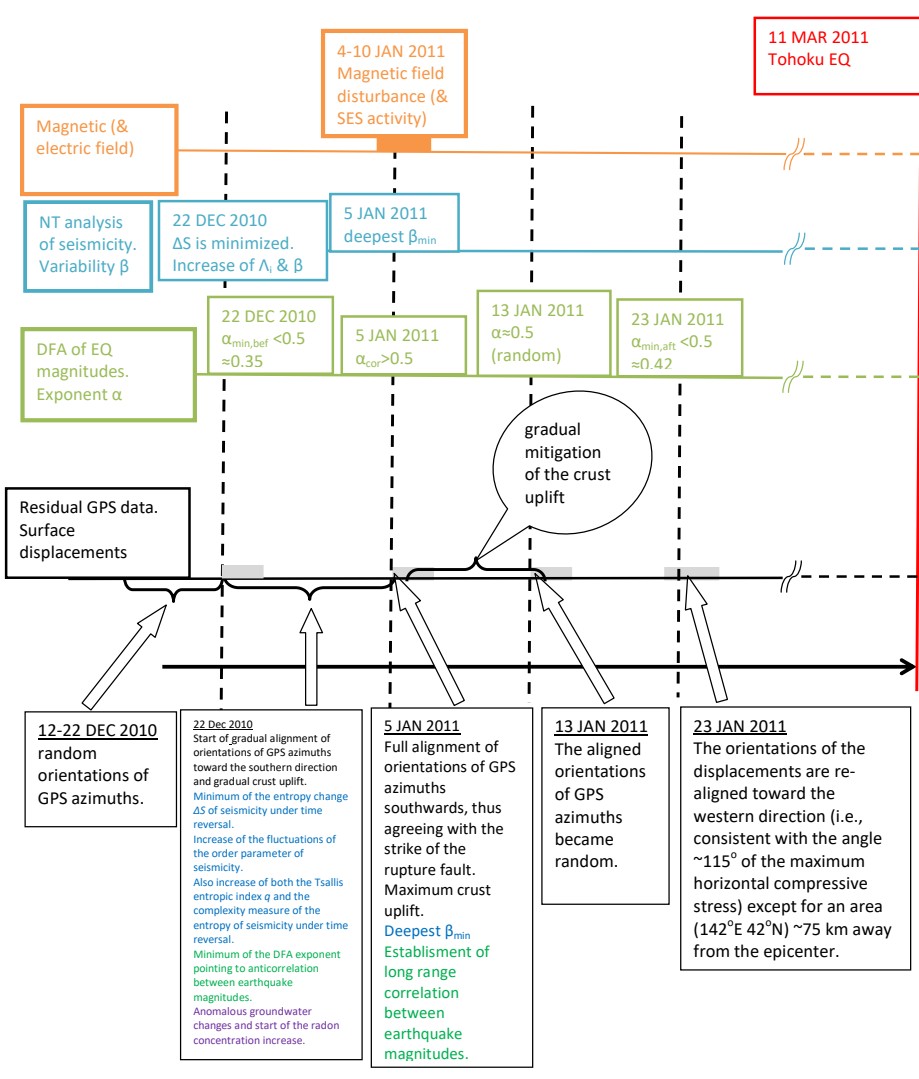

**Figure 3.** Schematic diagram that compiles the multidisciplinary changes before the Tohoku EQ. Period of observations: from 12 December 2010 until 23 January 2011. The black letters describe the observations by means of residual GPS data, the orange by Earth's magnetic field variations, green obtained by DFA, blue by natural time analysis and purple by anomalous groundwater and radon changes.