# Peer review of "Phenomena preceding major earthquakes interconnected through a physical model"

_Annales Geophysicae, 2019_

## Referee Comment (RC1) · Anonymous Referee #1 · 28 Mar 2019

This paper treated with a number of precursory phenomena before the Tohoku M9 earthquake, systematically. The authors divided observed precursors into two categories which correspond to two different physical stages. It is very interesting.

In the beginning, the word "order parameter" suddenly appeared. As far as the reviewer's knowledge, "order parameter" is a common word for a physicist, but not for a seismologist. It means that the authors are considering earthquake is a kind of the phase transition process. It is better to explain a little bit more. Because, in the future, seismologists' understanding becomes very important in this approach.

In any case, the sentences are redundant. Can you shorten the text more? The reviewer feels that expression is old-fashioned. More simple wording would be better.

[Figure]

Minor comments: P2 L26 5 1/2 is too precise. Say 6months is OK?

P8 L10 the initiation of the anomalous Earth's magnetic field variations and the minimum $\beta$min of the fluctuations of the order parameter of seismicity. <- $\beta$ varies case by case. You cannot say the exact date of January 5. The expression is too strong.

P9 L26 Anomalous magnetic field variations started (which reflects the initiation of a strong SES activity). -> In the case of Tohoku M9EQ, no electric field observation. The expression is too strong.

In figure 3: Pale colors are not easy to read. Please change some colors.

---

## Referee Comment (RC2) · Anonymous Referee #2 · 30 Mar 2019

The article "Phenomena preceding major earthquakes interconnected through a physical model" by Varotsos, Sarlis & Skordas, deals with a very interesting topic. The article provides a short review of the precursors to the M9 Tohoku earthquake that have been reported during a $\sim$1 month long time period (22 Dec. 2010 – 23 Jan. 2011), i.e. $\sim$2.5 to $\sim$1.5 months before the Tohoku earthquake of 11 Mar. 2011 and examines whether these precursors are in agreement with the pressure stimulated polarization currents (PSPC) condensed matter physical model, which has been proposed for the generation of the Seismic Electric Signals (SES). The authors show that all these precursors are directly evidenced from the PSPC model, except probably of the anomalous changes of the level and temperature of confined groundwater.

The article is well written and easy to follow. It offers a novel important unifying view

about a number of multidisciplinary precursory phenomena. Thus, in my opinion, merits publication in Annales Geophysicae. A few minor improvements can be made as follows:

In p.7, line 7, the authors should check whether the subscript "bef" is correct or "aft" should be used instead.

Figure 3 is a very informative figure. However, the authors should be careful so that the text is fairly readable. The second from left text box that appears at the bottom uses too small fonts. In my opinion, the authors should increase the font size at least for the specific text box, or they can try to move part of the text to the figure legend.

---

## Author Comment (AC1) · 16 Apr 2019

First we include in brackets the points raised by the Reviewer in italics and our answer follows in plain text.

[*In the beginning, the word "order parameter" suddenly appeared. As far as the reviewer's knowledge, "order parameter" is a common word for a physicist, but not for a seismologist. It means that the authors are considering earthquake is a kind of the phase transition process. It is better to explain a little bit more. Because, in the future, seismologists' understanding becomes very important in this approach.*]

We will explain this point as to become more helpful to the readers.

[*In any case, the sentences are redundant. Can you shorten the text more? The*

[Figure]

*reviewer feels that expression is old-fashioned. More simple wording would be better.*]

We will shorten the text accordingly.

[*P2 L26 5 1/2 is too precise. Say 6months is OK?*]

We will rephrase it in the revised version.

[ *P8 L10 the initiation of the anomalous Earth's magnetic field variations and the minimum $\beta_{\min}$ of the fluctuations of the order parameter of seismicity. <- $\beta$ varies case by case. You cannot say the exact date of January 5. The expression is too strong.*]

We will rephrase it appropriately in the revised version using "around January 5"

[*P9 L26 Anomalous magnetic field variations started (which reflects the initiation of a strong SES activity). -> In the case of Tohoku M9EQ, no electric field observation. The expression is too strong.*]

We will rephrase it appropriately in the revised version

[*In figure 3: Pale colors are not easy to read. Please change some colors.*]

Yes, we will redraw the figure by changing the colors in order to become more clear for the readers.

We would like to thank the Reviewer for his/her constructive review.

---

## Author Comment (AC2) · 16 Apr 2019

First we include in brackets the points raised by the Reviewer in italics and our answer follows in plain text.

[*In p.7, line 7, the authors should check whether the subscript "bef" is correct or "aft" should be used instead.*]

Yes, the reviewer is right. We will change "bef" to "aft" in the revised version of the manuscript.

[*Figure 3 is a very informative figure. However, the authors should be careful so that the text is fairly readable. The second from left text box that appears at the bottom uses too small fonts. In my opinion, the authors should increase the font size at least for the*

[Figure]

*specific text box, or they can try to move part of the text to the figure legend.*]

We will redraw the figure by changing the colors (as was also requested by another Reviewer) and by increasing the fonts in order to become more clear for the readers.

We would like to thank the Reviewer for his/her constructive review.